# Computational modeling of color perception with biologically plausible spiking neural networks

**Hadar Cohen-Duwek**[1], **Hamutal Slovin**[2], **Elishai Ezra Tsur**[1]\*

**1** Neuro-Biomorphic Engineering Lab, Department of Mathematics and Computer Science, The Open University of Israel, Ra'anana, Israel, **2** The Gonda Multidisciplinary Brain Research Center, Bar-Ilan University, Ramat Gan, Israel

\* elishai@nbel-lab.com

## Abstract

Biologically plausible computational modeling of visual perception has the potential to link high-level visual experiences to their underlying neurons' spiking dynamic. In this work, we propose a neuromorphic (brain-inspired) Spiking Neural Network (SNN)-driven model for the reconstruction of colorful images from retinal inputs. We compared our results to experimentally obtained V1 neuronal activity maps in a macaque monkey using voltage-sensitive dye imaging and used the model to demonstrate and critically explore color constancy, color assimilation, and ambiguous color perception. Our parametric implementation allows critical evaluation of visual phenomena in a single biologically plausible computational framework. It uses a parametrized combination of high and low pass image filtering and SNN-based filling-in Poisson processes to provide adequate color image perception while accounting for differences in individual perception.

## Author summary

In this work, we propose a biologically plausible computational framework for color perception. The model initiates with simulating the responses of single and double opponent cells to a visual stimulus in chromatic and achromatic channels. The double opponent and the intensity channels are reconstructed using spiking neural networks and linearly combined with the single opponent channels to provide the perceived image. Our model allows the attribution of perceptual differences to the proportions between the single and double opponent cells' activity, while being general enough to account for a wide range of visual phenomena including color constancy, color assimilation, and ambiguous color perception.

## Introduction

One of the most fundamental challenges in modeling human cognition is linking high-level experiences to low-level biologically plausible computational models. Advances in

**Data Availability Statement:** The project's Github with the complete codebase available in: https://github.com/NBELab/PLoS_Comp_2022.

**Funding:** This work was supported by The Open University of Israel Research Grant, which was

granted to E.E.T. E.E.T. is a senior faculty member at The Open University of Israel. The funders had no role in study design, data collection and analysis, decision to publish, or preparation of the manuscript.

**Competing interests:** The authors have declared that no competing interests exist.

computational neuroscience, cognitive science, and artificial intelligence continually power our attempts to shed light on this grand challenge. One of the most interesting aspects of human cognition is visual perception. Visual perception initiates with the derivation of light intensity and color by retinal circuitry, which is propagated to the Lateral Geniculate Nucleus (LGN), finally advancing to the primary visual cortex (V1) and on to higher processing areas [1]. Interestingly, while visual information is represented as Spatio-temporal edges, the perceived field of view features complete colorful filled-in surfaces, indicating that the brain reconstructs visual constructs from edges [2]. Fronting extensive empirical research, two prominent theories have been suggested to govern perceptual filling-in: (1) **Symbolic or cognitive theory** according to surfaces' color and shape are represented in higher- areas of visual processing; and the (2) **Isomorphic theory**, according to surfaces emerge from activation spreads from edges to the centers across the retinotopic map. This activation pattern propagates across a two-dimensional grid of neurons, representing a planar field of view. The underlying neural mechanism of perceptual filling-in remains unclear, as experimental evidence supports both hypotheses [3]. In visual perception modeling, chromatic and achromatic receptive fields are typically modeled using spatial derivatives kernels [4]. Recently, we proposed biologically plausible Poisson-driven perceptual filling-in Spiking Neural Networks (SNN), demonstrating the reconstruction of images from their gradients [5]. SNNs are considered biologically plausible as they feature spiking neurons and local learning rules without a Central Processing Unit (CPU) nor a register-based memory.

In V1, visual data is represented as Spatio-temporal edges by color-responsive single- and double-opponent neurons. While single opponent cells merely report the color of their receptive field, double-opponent report chromatic edges and are orientation-selective [6–10]. Both single and double opponent neurons were hypothesized to govern color perception. Recently, Shapely and Colleagues suggested that while single-opponent neurons play a vital role as spatial integrators at static low color contrast visual scenes, at higher contrast, and where colors dynamically change, double-opponent neurons govern perception [7]. Visual perception also comprises various processing pathways, combining chromatic and achromatic edge processing. While the achromatic pathway reports on color-oblivious edges, the chromatic pathway combines Red/Green and Yellow/Blue edges.

In this work, we extend our previous model, proposing an isomorphic theory-driven biologically plausible SNN for the reconstruction of colorful images from retinal inputs (**Fig 1**). We introduced a colored image (stimulus) to chromatic and achromatic channels, comprising models of single and double opponent neurons. The derived chromatic and achromatic edges were introduced into recurrent SNNs, implementing evidence-based feedback (horizontal) connections [11,12] to reconstruct the embedded surfaces. Finally, the resulting surfaces were linearly combined with single opponent outputs to produce a perceived image. A weighting scheme controls the dominance of each channel in the perceived image, as was described by Shapley and colleagues [7].

We further used our model to demonstrate and critically explore three important visual phenomena: (1) **color constancy**, in which an object' perceived color is perceived under varying lighting conditions [13]; (2) **the color assimilation grid illusion,** in which the color of a grid is assimilated into the underlying black and white surfaces; and (3) **ambiguous color perception** (e.g., #TheDress and #TheShoe). Interestingly, perceptual filling-in-driven visual illusions, featuring chromatic and achromatic phenomena, have been long known for shedding new light on neural mechanisms in the visual system [14–19]. For example, extensive research has been conducted on color constancy [20], deciphering it as a result of either high-level processing with which color is estimated in accordance with prior experience [21,22], or low-level retinal [23] and V1 [24–26] processing. Our work provides a unique biological plausible

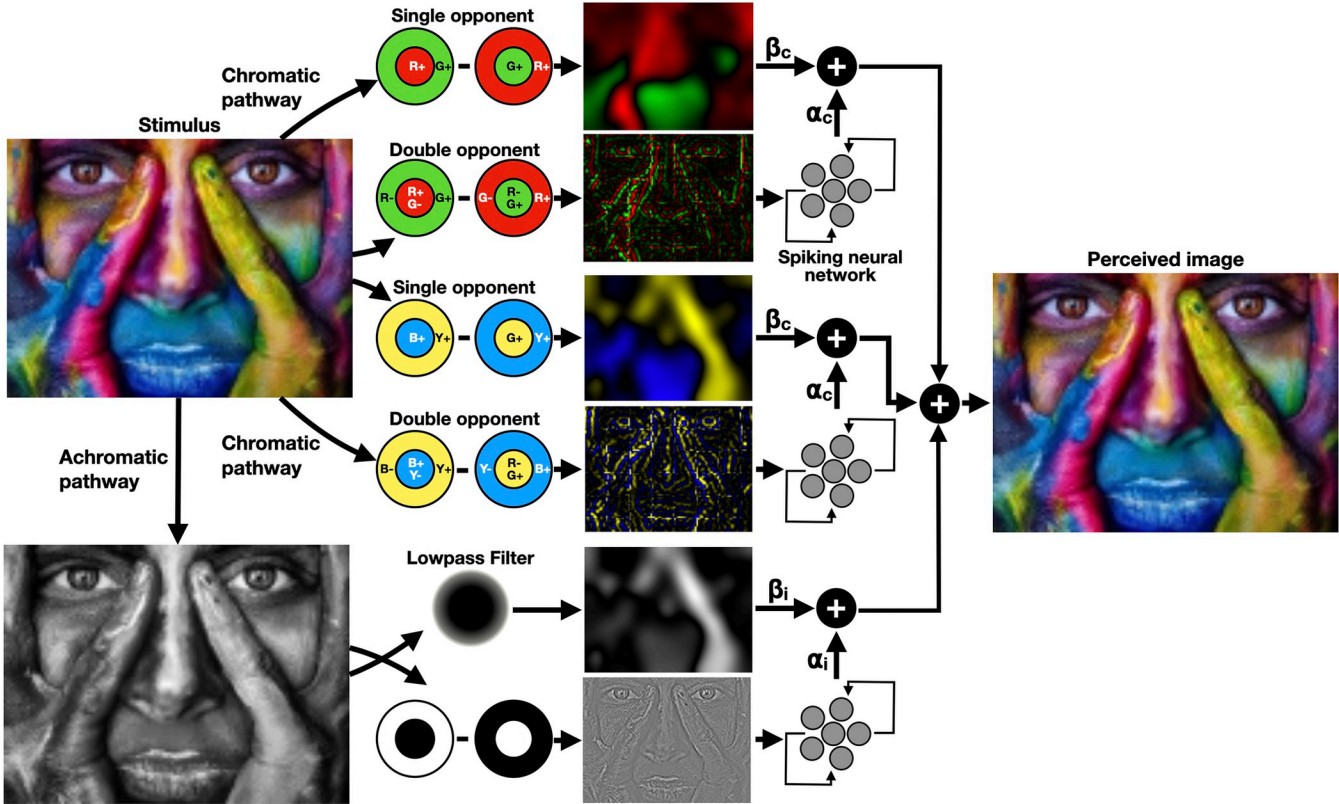

**Fig 1. A biological plausible computational framework for color perception.** The model initiates with simulating the responses of single and double opponent cells to a visual stimulus in chromatic (red-green and blue-yellow) and achromatic channels. The DO and the intensity channels are reconstructed using SNNs and linearly combined with the SO channels to provide the perceived image. Image by Alexander Ivanov (Pixabay).

computational framework in which these intricate visual phenomena can be critically and exploratory examined.

## Methods

### Ethics statement

All experimental procedures were approved by the Animal Care and Use Guidelines Committee of Bar-Ilan University, supervised by the Israeli authorities for animal experiments, and conformed to the National Institutes of Health (NIH) guidelines.

### The neural engineering framework

SNNs comprise a network of interconnected spiking neurons [27]. In this work, we utilized the Neural Engineering Framework (NEF) [28], a theoretical framework with which spiking neurons can be used to design functional large-scale neural networks. With NEF, numerical high-dimensional constructs (e.g., vectors and functions) can be loosely encoded, decoded, and transformed. Following NEF, a spikes train $\delta$ is defined using:

$$\delta_i(\mathrm{x}) = G_i[\alpha_i e_i \mathrm{x} + J_i^b], \qquad (1)$$

where $i$ is the neuron identifying index, x is the stimulus, $e$ is the neuron's preferred stimulus

(encoding vector), $G$ is a spiking neuron model (e.g., Leakey Integrate and Fire (LIF) neuron model), $\alpha$ is a gain term, and $J^b$ is a fixed background current.

An encoded high-dimensional numerical construct (vector), can be linearly decoded as $\hat{x}$ using:

$$\hat{x} = \sum_i^N a_i(x)d_i, \tag{2}$$

where $N$ is the number of spiking neurons, $a_i(x)$ is the postsynaptic low-pass filtered response of neuron $i$ to stimulus x and $d_i$ is a representational decoder. Representational decoders are optimized to reconstruct x using least squared optimization. Eqs 1 and 2 describe the encoding and decoding of vectors with neural spiking activity within neuronal ensembles. Propagation of data from one ensemble to another can be realized through weighted synaptic connections (transformational decoders). Transformational decoders can be optimized such as $x$ could be transformed to an arbitrary $f(x)$. Dynamic behavior is realized by recurrently connecting neuronal ensembles (thus, integrating NEF's representation and transformation principles). NEF can be used to resolve the dynamic:

$$^dx/_dt = f(x(t)) + u(t), \tag{3}$$

where $u(t)$ is input from another neural ensemble, defining a recursive connection that resolves the transformation: $\tau \cdot f(x)+x$, where $\tau$ is the synaptic time constant. A detailed description of NEF is available in [28].

## Single and double opponent channels

Our model initiates by simulating the responses of single and double opponent cells to a visual stimulus [29] (**Fig 1**). We followed the central dogma in which the visual system utilizes separate channels for processing achromatic data and colors of different wavelengths [9]. For the chromatic pathway, we have implemented two pathways: $L/M$, and $(L+M)/S$, where $L$ represents a long light wavelength (red), $M$ represents an intermediate light wavelength (green), and $S$ represents a short light wavelength (blue). We used the RGB channels of the input image to describe the $L$, $M$, and $S$ color intensities. The achromatic pathway comprises a Low Pass Filter (LPF) and a derivative kernel (following the on-center—off-center receptive fields of retinal ganglion cells).

The three RGB color channels of the visual stimulus were converted to two red-green $SO_{RG}$ and blue-yellow $SO_{BY}$ single opponent channels, and the achromatic (grayscale) channel, denoted $I_{LPF}$, using:

$$SO_{RG} = RG * G(x, y, \sigma), \tag{4}$$

$$SO_{BY} = BY * G(x, y, \sigma), \tag{5}$$

$$I_{LPF} = I * G(x, y, \sigma), \tag{6}$$

where $G(x,y,\sigma)$ is the normalized Gaussian Kernel: $\frac{1}{N} e^{-\frac{(x^2+y^2)}{2\sigma^2}}$ in which $N = \sum_{x \in s}\sum_{y \in s} e^{-\frac{(x^2+y^2)}{2\sigma^2}}$, '*' denotes the convolution operator, simulating the low-pass properties of the single opponent channel [7], [8], [30]; and $RG$, $BY$ (the chromatic channels), and $I$ (the achromatic channel)

were defined using:

$$\begin{pmatrix} RG \\ BY \\ I \end{pmatrix} = M_{opp} \begin{pmatrix} R \\ G \\ B \end{pmatrix} = \begin{pmatrix} \frac{1}{\sqrt{2}} & -\frac{1}{\sqrt{2}} & 0 \\ \frac{1}{\sqrt{6}} & \frac{1}{\sqrt{6}} & -\frac{2}{\sqrt{6}} \\ a & b & c \end{pmatrix} \begin{pmatrix} R \\ G \\ B \end{pmatrix}, \tag{7}$$

where $M_{opp}$ is the color opponent transformation matrix in which $a = 0.2989$, $b = 0587$, and $c = 0.114$. The spatial dimension $s$ of the gaussian kernel (measured in pixels) is $\left\{ -\lfloor \frac{W}{2} \rfloor, \ldots, \lfloor \frac{W}{2} \rfloor \right\}$, where $W$ was set to 21 during model execution and to 11 or 21 during parameter evaluation (see parameter evaluation below for further details).

The chromatic double-opponent channels: $DO_{RG}$ and $DO_{BY}$ as well as the achromatic derivative signal $I_{on-off}$ were derived by convolving each chromatic single opponent channel with the Discrete Laplacian operator $L = \begin{pmatrix} 0 & -1 & 0 \\ -1 & 4 & -1 \\ 0 & -1 & 0 \end{pmatrix}$, constituting:

$$DO_{RG} = RG * L \approx \Delta RG, \tag{8}$$

$$DO_{BY} = BY * L \approx \Delta BY, \tag{9}$$

$$I_{on-off} = I * L \approx \Delta I. \tag{10}$$

Following Fig 1, $I_{on-off}$, $DO_{RG}$ and $DO_{BY}$, $SO_{RG}$, $SO_{BY}$ and $I_{LPF}$ were represented with spiking neurons using Eq 1 and introduced into SNNs for surface filling-in.

## Perceptual Filling-in with spiking neurons

In V1, visual data is represented as Spatio-temporal edges, constituting the image's gradients. The perception of filled surfaces from image gradients can be described using the diffusion/heat equation:

$$\frac{\partial I}{\partial t} - \Delta I(x, y) = div\left( \nabla I_{input} \right) \tag{11}$$

where $\nabla = \left[ \frac{\partial}{\partial x}, \frac{\partial}{\partial y} \right]$ is the gradient operator, $\Delta = \left[ \frac{\partial^2}{\partial x^2} + \frac{\partial^2}{\partial y^2} \right]$ is the Laplacian operator, $div$ is the divergence ($div\, \boldsymbol{F} = \frac{\partial F_x}{\partial x} + \frac{\partial F_y}{\partial y}$), $I$ is the perceived image (i.e., the reconstructed image), and $I_{input}$ is the input image (stimulus) [31], [32]. In the diffusion process, the inactive center of the V1-represented stimulus is gradually filled-in with neuronal activity, supporting the perception of light intensity at the center of the outlined stimulus. This diffusion-governed perceptual filling-in is often referred to as 'immediate'[2], following experimental evidence supporting the almost instantaneous reconstruction of a perceived image [17], [18], [33]. This fast dynamic allows the dismissal of $\frac{\partial I}{\partial t}$, the diffusion equation's dynamic phase. Eq 11 can be

therefore simplified to the steady-state Poisson equation:

$$\Delta I(x, y) = -div(\nabla I_{input}). \tag{12}$$

While the Poisson equation can be realized numerically by various techniques [34–36], we recently demonstrated a biologically plausible solution using NEF-defined recurrent SNNs [5]. Our recurrent SNN iteratively solves the Poisson equation by rearranging Eq 11, as: $\frac{\partial I}{\partial t} = div\left(\nabla I_{input}\right) + \Delta I$, which in conjunction with Eq 3, allows the definition of the recurrent connection:

$$feedback(I) = \tau \cdot (div(\nabla I_{input}) + \Delta I) + I. \tag{13}$$

Eq 13 can be iteratively defined as:

$$I_k = \tau \cdot (div(\nabla I_{input}) + \Delta I_{k-1}) + I_{k-1}. \tag{14}$$

Following Eq 14, the perceived image $I_k$ can be iteratively reconstructed in every timestep $k$. Eq 14 can be discretized by the Discrete Laplace operator $L$:

$$I_k = \tau \cdot (L(I_{input}) + L(I_{k-1})) + I_{k-1}. \tag{15}$$

Finally, the image $I(x,y)_k$ (the perceived pixel in $(x,y)$ at time step k) can be derived using:

$$I(x, y)_k = \tau \cdot (div(\nabla I(x, y)_{input}) + I(x, y - 1)_{k-1} + I(x, y + 1)_{k-1} + I(x - 1, y)_{k-1}$$
$$+ I(x + 1, y)_{k-1} - 4 \cdot I(x, y)_{k-1}) + I(x, y)_{k-1} \tag{16}$$

In Eq 16, each neuron has four recurrent connections with its four neighboring cells and one recurrent connection with itself. In each time step, neural activity is spread to his adjacent neurons. Here, we realized Eq 16 using a recurrently connected single-layer SNN. Therefore, this connectivity scheme can be referred to as a horizontal neural connection [11,12]. In this work, we further utilize this SNN to demonstrate color perception and the perception of visual artifacts.

## Image perception

For each color opponent channel as well as the intensity channel, the above filling-in process was applied separately. Eqs 17–19 describe the inputs for Eq 12. Solving each equation yields the filled-in surfaces $O_{RG}$, $O_{BY}$, $O_I$ for the RG, BY and Intensity opponent channels, respectively.

$$\Delta O_{RG}(x, y) = -DO_{RG} \tag{17}$$

$$\Delta O_{BY}(x, y) = -DO_{BY}. \tag{18}$$

$$\Delta O_I(x, y) = -I_{on-off}. \tag{19}$$

The perceived image is generated by combining the reconstructed achromatic pathways $P_I$, with the single and double opponent channels in each color pathway ($P_{RG}$, $P_{BY}$):

$$P_{RG} = \beta_c \cdot SO_{RG} + \alpha_c \cdot O_{RG}, \tag{20}$$

$$P_{BY} = \beta_c \cdot SO_{BY} + \alpha_c \cdot O_{BY}, \tag{21}$$

$$P_I = \beta_i \cdot LP_I + \alpha_i \cdot O_I, \tag{22}$$

where $P_{RG}$, $P_{BY}$ are the perceived results of the red-green, and blue-yellow channels, respectively; $\alpha_c$, $\beta_c$, $\alpha_i$ and $\beta_i$ are weight parameters, indicating the weighted contribution of each channel to the perceived result. In this study, we defined $\beta = 1 - \alpha$, allowing us to solely control $\alpha$ in the simulations.

To transform the perceived result to an RGB image, the three opponent channels are converted back to RGB representation using the inverse opponent transformation:

$$\begin{pmatrix} R \\ G \\ B \end{pmatrix} = \{M_{OPP}\}^{-1} \begin{pmatrix} P_{RG} \\ P_{BY} \\ P_I \end{pmatrix}. \tag{23}$$

We used the Learning Perceptual Image Patch Similarity (LPIPS) metric to measure the perceptual distance between the visual stimulus and the reconstructed image. In LPIPS, deep visual features are extracted from pairs of images derived from ImageNet-trained neural networks [37] and compared using a weighted L2 distance (Euclidean distance). Weights were adjusted such that this similarity measure agrees with human perceptions of patch similarity [38], based on the Berkeley Adobe Perceptual Patch Similarity (BAPPS) dataset. BAPPS contains two-alternative-force-choice (2AFC) and just-noticeable-difference (JND) judgment experiments. As part of the 2AFC experiment, two distortions are applied to a reference image patch, and observers must choose which distortion is closest to the original. In the JND experiment, the observer is asked to determine if two patches—one reference and one distorted—are the same or different.

The LPIPS distance is defined as:

$$d(O, P) = \frac{1}{N} \sum_{i=1}^{N} \sum_l \| w_l \cdot (\Phi_l(O^i) - \Phi_l(P^i)) \|_2^2, \tag{24}$$

where $O^i$ and $P^i$ are the RGB values of pixel $i$ in the original and the predicted (reconstructed) image, respectively; $N$ is the number of pixels, and $\Phi_l(\cdot)$ donates the feature activations at the $l$-th layer of the AlexNet [39] network $\Phi$. Weights $w_l$ were optimized using the Berkeley Adobe Perceptual Patch Similarity (BAPPS) dataset to match human perception; Perceptual distance $d$ was calculated by using the first five layers of AlexNet.

## Model simulation

To evaluate our SNNs-driven model for color perception, we implemented the model using the Nengo neural compiler (implemented with Python), with which high-level descriptions can be translated to low-level spiking neurons [40]. The model was directly introduced with the single and double opponent cells (*SO* and *DO*, Eqs 4–6,8–10) derived from RGB images. For the *DO* cells, the spatial extents of the filters were used to represent a high-pass filter (Laplacian), and for the *SO* cells, we chose a low-pass filter with a relatively large support (wide spatial Gaussian profile). In simulations, the spatial parameters of the Gaussian kernel were $W = 21$ pixels and $\sigma = 5$ (Eqs 4–6). Each pixel was encoded with five ensembles, each constituting 20 spiking neurons, representing five channels ($SO_{RG}$, $SO_{BY}$, $DO_{RG}$, $DO_{BY}$ and $I_{\mathrm{on-off}}$). Time constant $\tau$ (Eq 16) was set to 0.25 in all simulations. Neurons were defined with a Spiking-Rectified-Linear activation function [41]. Simulations were accelerated on a 12GB NVIDIA Tesla K80 GPU using the OpenCL-based Nengo Simulator [40].

## Voltage-sensitive dye imaging and analysis

Imaging and experimental procedures were fully described in [42]. Briefly, a monkey (6 years old 13 kg male macaque monkey (*Macaca fascicularis*)) was trained on a fixation task while presented (21-inch CRT monitor; 85 Hz refresh rate; 100 cm from the monkey's eyes) with black (CIE-xy = 0.279, 0.266) or red (CIE-xy = 0.616, 0.341) squared surfaces of equal luminance (15.5 cd/m$^2$), background (CIE-xy = (0.279, 0.28); luminance (7.3 cd/m$^2$) and a variable size. We used a 3 to 4 seconds prestimulus (varied randomly) and a 300 ms stimulus time. The center positions of all surfaces in the visual field were identical (stimulus fixation within 2˚ about the fixation point; verified using eye movement monitoring). The monkey was anesthetized, ventilated, and anchored (cemented to the cranium with dental acrylic) using two 25mm cranial windows, bilaterally placed over the primary visual cortices. The visual cortex was exposed (3–6 mm anterior to the Lunate sulcus) and stained using Oxonol voltage-sensitive dyes. We used Micam Ultima's imaging system, providing a resolution of 10$^4$ pixels at 10 kHz, each pixel summing the neural activity from about 500 neurons, located at the upper 400 μm of the cortex. VSDI maps were averaged at 60–100 ms after stimulus onset. We computed spatial cuts crossing through the edges and center of the activation patches (an illustration of the spatial profile for the 1˚ square is shown in **Fig 2B** (top). VSDI responses (**Fig 2A**) were averaged over the width of the spatial cuts resulting in the spatial activity profiles shown in **Fig 2B**.

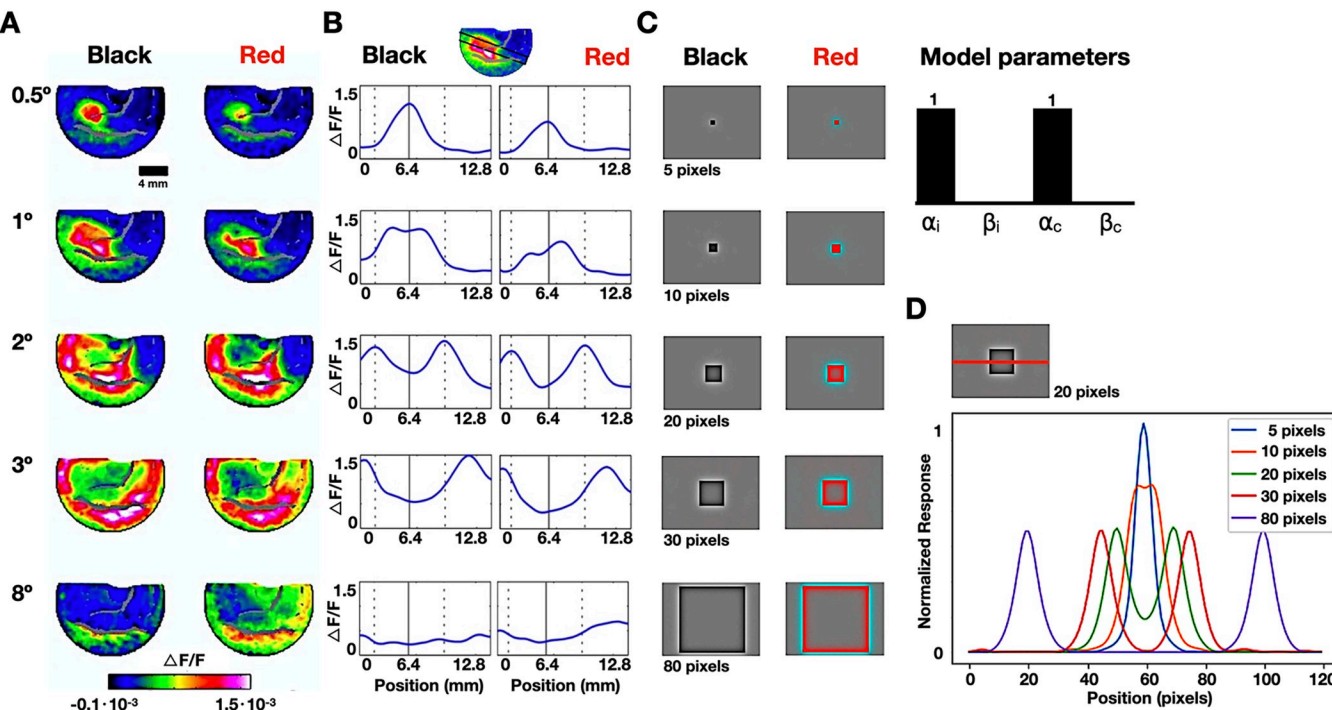

**Fig 2. V1 recording and simulation results during perceptual filling-in of black and red squared surfaces.** A. Averaged early (60–100 ms following stimulus onset) Macaque V1 VSDI-measured neural activity map following exposure to black (left) and red (right) squared surfaces with various sizes (0.5˚– 8˚) (see Methods); B. Spatial profiles crossing through the edges and center of the V1 activation patches. The continuous vertical line marks the peak activation position in the 0.5˚ square response profile, which corresponds to the center of square in larger stimuli. Responses to a 2˚ square are marked with vertical dashed lines; C. Model-derived results for the reconstruction of black and red squared surfaces with various sizes squares; D. Cross-sectional profiles of each reconstructed square along the x-axis.

## Results

### Filling-in V1 recording and simulation

Macaque V1 neural responses to visual stimuli of black and red squared surfaces in varying sizes, ranging from 0.5˚ to 8˚ visual degrees, were recorded using VSDI (originally reported in [42]) (**Fig 2A**). We found that the spatial V1 neural responses pattern for small surfaces (0.5˚ and 1˚) were 'filled-in', corresponding to the stimulus topographic map. Neural responses for larger surfaces (2˚ to 8˚) showed 'un-filled' areas, indicated by the low response amplitude at the surface's center. Furthermore, we derived spatial cross-sectional measurements through the edges and center of the activation patches (an illustration of the spatial profile for the 1˚ square is shown in **Fig 2B**, top). VSDI response was averaged over spatial cross-sections resulting in an activity profile, depicted in **Fig 2B**. For comparison, we used our image reconstruction model (**Fig 1**) to reconstruct five squares ranging from 5 to 80 pixels in length, each corresponding to a different visual modality used in the experimental setting (**Fig 2C**). Since VSDI recorded signals from the 2nd and 3rd V1 layers, which are double opponent cells dominant [6], we set $\alpha_i$ and $\alpha_c$ to 1 for reconstruction. Briefly, these cortical layers are imaged using VSDI at high spatial resolution (mesoscale, 502 $\mu m^2$/pixel) and temporal resolution (100 Hz). While VSDI acquired signals emphasize subthreshold membrane potentials, it reflects supra-threshold membrane potentials (i.e., spiking activity). The main advantage of this technique is the combination of wide-field imaging with high spatio-temporal resolution, enabling the visualization of the whole cortical activity patterns evoked by a visual stimulus. As a result, in this section, the simulation solely considers the double opponent cells channel. In the reconstructed surfaces, only the small 5- and 10-pixels squares were completely filled-in. The centers of the larger reconstructed squares were unfilled, corresponding to the neuronal inactive patches shown in VSDI. We measured the spatial activation profiles across the reconstructed squares (along the x-axis) and applied a 1D Gaussian filter with $\sigma = 2$ to smooth the responses (**Fig 2D**). Our cross-sectional simulation results correspond to the experimental neuronal activation profiles we experimentally obtained in V1. We further compared the reconstructed VSDI profiles with the recorded VSDI profiles using linear regression. The stimulus edges in each VSDI profile were aligned (across the x-axis of each profile) to its corresponding simulated profile and $R^2$ and its corresponding p-value were derived (**S1 Fig**). Results shows $R^2$ of 0.974, 0.824, 0.575, 0.938 and p-values of $1.3 \cdot 10^{-55}$, $2.5 \cdot 10^{-27}$, $4.7 \cdot 10^{-14}$, $3.3 \cdot 10^{-45}$ for the 0.5˚, 1˚, 2˚ and 3˚ profiles, indicating a good model fit. $R^2$ was not calculated in the 8˚ profile since the signal was essentially noise.

### Color perception

We evaluated our color perception model by reconstructing four color images: a photograph of a colored face, an image of a building with reflective surfaces, a dimmed lighted photograph of the Louvre Museum, and a synthetic red square. The resulting reconstructions with various values of $\alpha_i$, $\beta_i$, $\alpha_c$, and $\beta_c$, are shown in **Fig 3**. As a general guideline, when $\alpha$ is high, and $\beta$ is small, the reconstructed colors and intensities demonstrate high pass filtering as high spatial frequencies are mostly reconstructed. For example, when $\alpha_i$ is high, the reflection of the Louvre in the water is clearer as the image's finer details are better exposed. Increasing $\beta$ (and reducing $\alpha$) instigates saturated colors and blurry edges. The synthetic red square image provides an intuitive illustration of this high- and low-pass filtering balance. When $\alpha_c = 1$, the red square is not entirely filled, and its color edges are enhanced. Since the model exhibits high-pass color filtration, the image's complementary color–cyan–appears at the square's exterior edges. As $\alpha_c$ decreases, the square's center is filled with a reddish hue. Interestingly, the reconstructed images are most similar to the original images when $\alpha_i$ and $\alpha_c$ are 0.5 indicating the

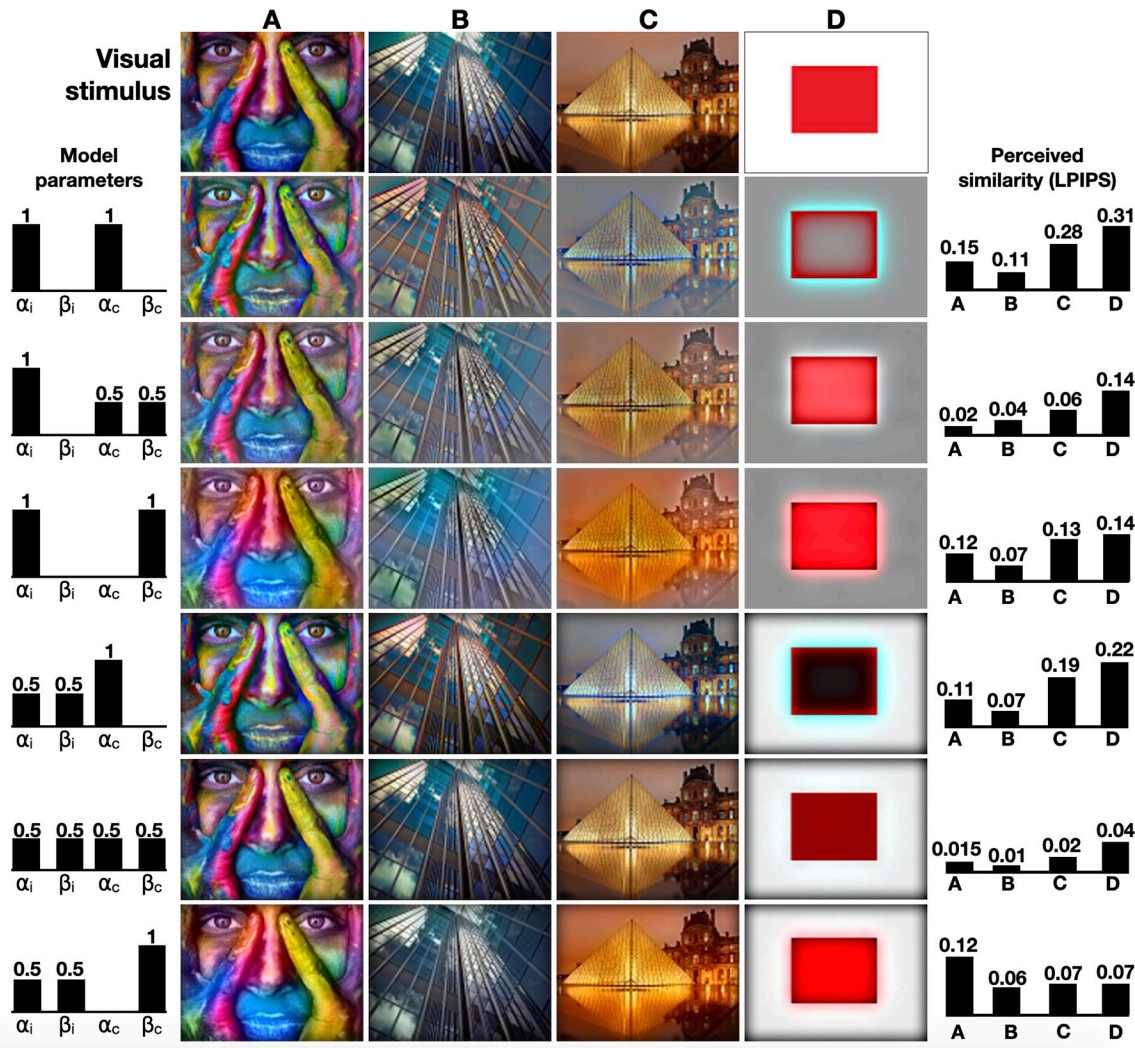

**Fig 3. Image reconstruction.** Visual stimuli are presented in the first row and the reconstructed images are shown below, generated with various values of $\alpha_c$, $\beta_c$, $\alpha_i$ and $\beta_i$ (aligned to the left of the corresponding images). LPIPS scores are shown on the right.

important contribution of the different color channels to adequate image perception. We evaluated the importance of the filling-in component by removing it from the proposed perceptual pipeline. Without a recurrent connection between the SO and DO channels, our model is simplified to a combination of low- and high-pass filters where most of the band-pass signals (intermediate frequencies) are absent. When $\alpha_i$ and $\alpha_c$ equal 1, the resulted reconstruction is a high-pass filter of the image (the image's Laplacian), whereas decreasing alphas merely adds low frequencies to the results (**S2 Fig**).

The model's reconstructed and original images were further compared using the LPIPS distance (**Fig 3**, right). As expected from visual inspection, the LPIPS distances for all four images were found lowest when $\alpha_i = 0.5$ and $\alpha_c = 0.5$, indicating the importance of multiple channel integration. Finally, we further evaluated the proposed model with a non-spiking version (a conventional neural network). A SNN is considered biologically plausible as is it uses spikes to represent and transform data through local learning rules (**Eq 2**). However, we can analytically solve the mathematical transformation our SNN strives to approximate. Under visual inspection, the reconstructed results of the spiking and non-spiking neural networks are similar, pointing out the

capacity of our model to exhibit relevant neural approximations. Interestingly, when measuring LPIPS distances during models' convergence, the SNN outperform the conventional neural network, consistently reporting lower distances (S3 Fig). This might be due to the noise-introductory effect, which is inherit in SNNs being a neural approximation, to the resolved diffusion process (Eq 16). While here we used a SNN to increase the biological plausibility of the model, the improved diffusive filling-in process may be an interesting topic for future work.

## Color constancy

Color constancy lies at the foundation of numerous visual illusions [20], [43]. In this section, we used the cube illusion created by Beau Lotto [44] to demonstrate how our SNN-driven biological plausible model can predict perceived colors under different illumination, as well as filter ambient illumination. The first row in Fig 4A illustrates three variations of the cube

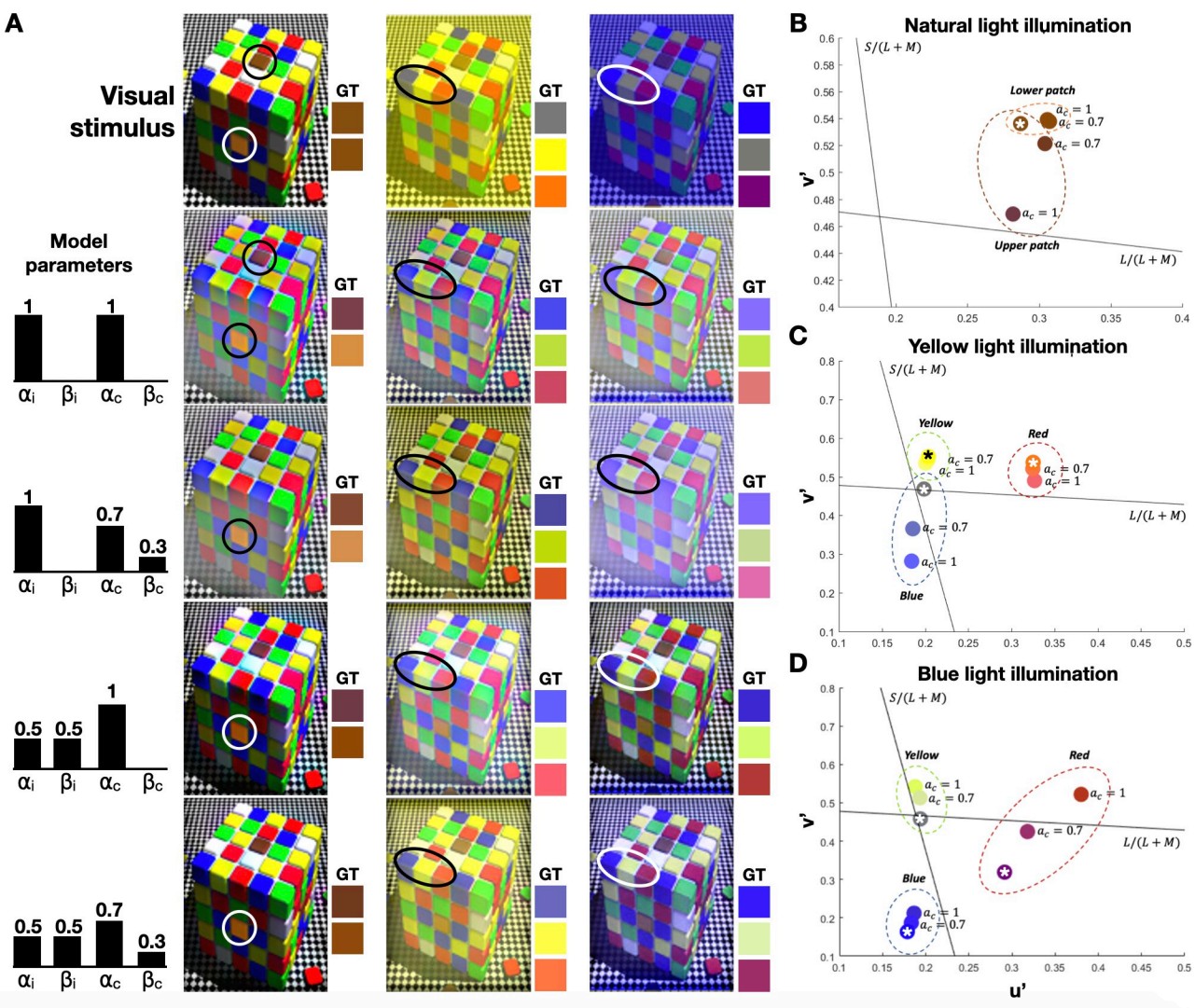

**Fig 4. Reconstruction of the cube illusion.** A. The original cube images under three illuminations: natural, yellow, and blue (First row). The model predictions with different sets of chromatic and achromatic parameters are shown in rows 2–5; B. Comparison between the true color (marked with an asterisk) and the predictions of the model with $\alpha_c = 1$ and $\alpha_c = 0.7$. Results are presented in u'v' (CIELu'v') color space. Each color circle surrounds the true and predicted colors of a sampled pixel in the patch. Black lines represent cone-opponent axes, S/(L + M) and L/(L + M). The intersection of the lines represents the achromatic point.

illusion, illuminated by natural, yellow, and violet\bluish lights (**Fig 4A,** left to right). When illuminated by natural (or white) light, the perceived color of each of the two marked patches (**Fig 4A**, left cube) is profoundly different, despite having the same color (ground truth; GT). A similar disparity between the perceived and GT colors is also apparent when yellow, and violet\bluish illuminations are used (**Fig 4A**, middle and right cubes). We reconstructed these images with our model with various values of $\alpha_i$, $\beta_i$, $\alpha_c$, and $\beta_c$ (**Fig 4A,** 2nd to 5th row). We were able to predict with our model the perceived color (i.e., the perceived and the GT colors are similar) under different illuminations. Furthermore, using different chromatic parameters, we could filter the ambient illumination in and out.

We further evaluated color constancy by observing the model's results in the perceptually uniform CIELu'v' color space [45]. We assessed the reconstructed cube illusion under natural light with different chromatic parameters ($\alpha_c = 1$ and $\alpha_c = 0.7$) and a constant achromatic alpha ($\alpha_i = 0.5$) (**Fig 4B**). While the patches' true colors are identical, we found that on the CIE-Lu'v' color space, when $\alpha_c = 1$, the predicted colors shift further away from each other as the upper patch becomes reddish and the lower patch's orange hue enhances. Under yellow illumination, while the yellow patch remains the same, when $\alpha_c = 1$, the blue shade of the blue patch's predicted color (gray in the original image) enhances, and the orange patch becomes reddish (**Fig 4C**). The predicted color of the cube under blueish illumination, when $\alpha_c = 1$, shows that while the blue patch remains the same, the gray patch becomes yellowish, and the purple patch becomes reddish (**Fig 4D**). Exploring the results of the yellowish and blueish illuminations (**Fig 4C and 4D**, respectively), we can see that the colors at $\alpha_c = 0.7$ are in between the original and the predicted colors at $\alpha_c = 1$.

## Color assimilation grid illusion

To demonstrate the importance of low-pass single opponent cells, we reconstructed the color assimilation grid illusion in which a selective colored grid is superimposed over an original grayscale image, resulting in a perceived color image [46]. The color assimilation grid illusion is demonstrated in **Fig 5** with a photograph of a colored face and a synthetic red square. We used two images representing two different grids' densities. Color assimilation is predominantly parameterized with line width (here, 3 pixels), line angle (here, 45˚), saturation ratio (here, 4), and line step, or the spacing between the grid's lines (here, 15 and 50 pixels). Images were rescaled to 90x120 and created using the "grid illusion" online tool [47]. Results show that when $\beta_c$ is high ($\beta_c = 1$), the predicted image's grey areas gained color, suggesting that the low-pass single-opponent part of the model must be dominant, allowing the assimilation grid illusion to take place. As $\beta_c$ value decreases, the low-pass effect of the single-opponent cells degrades, resulting in persistent achromatic areas. The model further demonstrates that, as expected, as the grid becomes denser, the perceived image gets further saturated with color, as testified by the measured LPIPS distances (**Fig 5, right**). We note that to compare the model's ability to reconstruct colors between the grid, as perceived in the illusion, LPIPS was calculated on the reconstructed and the full-color images (not the grid color images).

Interestingly, while the LPIPS distance for the red square is consistent with our perception (the most similar image is obtained when $\alpha_i = 0.5$ and $\alpha_c = 0$), the LPIPS distances for the colored face grid illusion are inconsistent. While the smallest LPIPS distances for the reconstructed colored face grid illusion are obtained when $\alpha_i = 1$ and $\alpha_c = 0.5$ (for both grid densities), by visual inspection, the perceived best results are obtained when $\alpha_i = 0.5$ and $\alpha_c = 0$. When presented with illusory and not natural images, our results demonstrate LPIPS's failure to measure perceptive similarity.

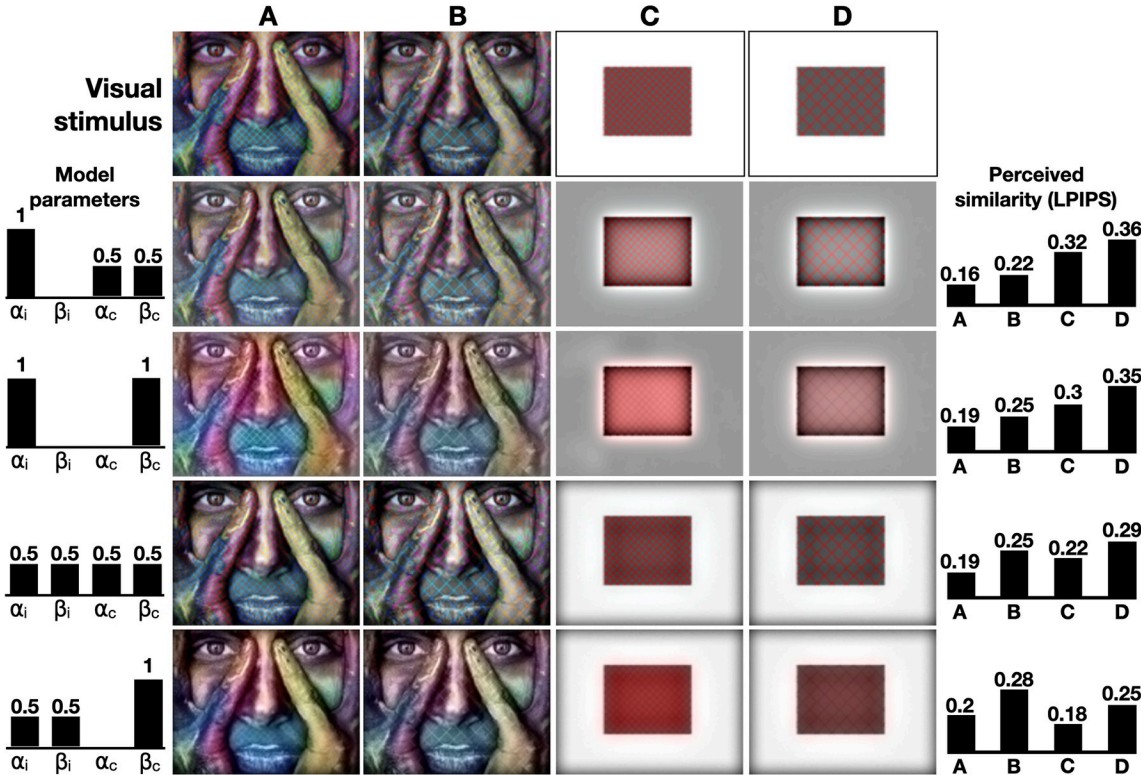

**Fig 5. Reconstruction of the color assimilation grid illusion.** A selective colored grid was overlaid on two grayscale images, creating the visual stimuli, shown in the first row. The model predictions with different sets of chromatic and achromatic parameters are shown in rows 2–5. LPIPS scores are shown on the right.

### #TheDress and #TheShoe

In 2015, two images, hashtagged on social media as #TheDress and #TheShoe, became viral as they depicted individual differences in color perception. In #TheDress image, some people perceived the dress's color as black and blue, while others perceived it as gold and silver (or gold and white) [48–50]. Similarly, while some perceived the colors of #TheShoe as pink and white, others perceived them as gray and cyan (turquoise) [50] (**Fig 6A**). Here we reconstructed these two famous photos, allowing us to examine the model's parameter space on the predicted colors (**Fig 6B**). In #TheDress reconstruction, results show that silver (achromatic) and gold (brownish) are perceived by setting the chromatic alpha to 1 ($\alpha_c = 1$), and blueish and black are scented with a chromatic alpha of 0.5 ($\alpha_c = 0.5$). In #TheShow reconstruction, results show that pink and light gray (slightly Cyanish) are perceived with a chromatic alpha of 1 ($\alpha_c = 1$), whereas the blueish and gray (dark achromatic) are scented with a chromatic alpha of 0.5 ($\alpha_c = 0.5$).

We further evaluated these results in the CIELu'v' color space (**Fig 6C**). Results show that the predicted colors are based on the chromatic parameters. When $\alpha_c = 1$, the dark brown (or black) patch of #TheDress becomes more saturated and brownish-orange (goldish) in appearance. The blue color of #TheDress turns more achromatic as it gets closer to the achromatic point. #TheShoe's gray patch becomes more reddish (pink) as it goes toward the red axis, and the cyan (turquoise) patch becomes more achromatic as it moves toward the achromatic point. However, when $\alpha_c = 0.5$, the predicted colors are getting closer to the ground truth colors in both photos.

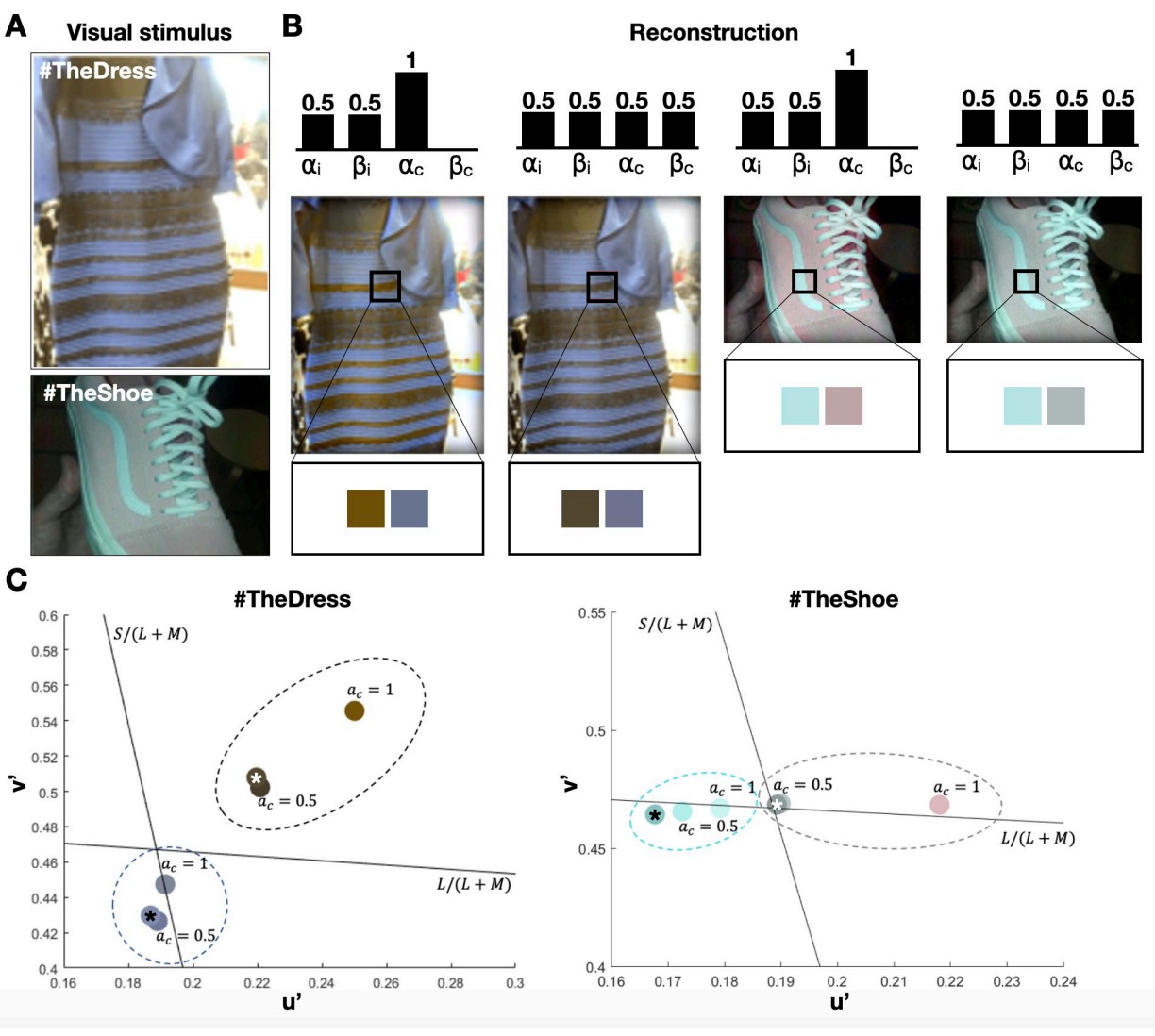

**Fig 6. Reconstruction of #TheDress and #theShoe photos.** A. The original images; B. Model's prediction with different sets of chromatic and achromatic parameters; C. Comparison between the true color (marked with an asterisk) and the predictions of the model Results are presented in u'v' (CIELu'v') color space. Each color circle surrounds the true and predicted colors of a sampled pixel in the patch. Black lines represent cone-opponent axes, S/(L + M) and L/(L + M). The intersection of the lines represents the achromatic point. **b) Comparison between the true color (mark with red \*) and the predictions of the model with $\alpha_c = 1$ and $\alpha_c = 0.5$ presented in u'v' (CIELu'v' 1976) color space. Each ellipse surrounds the true and the predicted colors of a sampled pixel in the patch. Black lines represent cone-opponent axes, S/(L + M) and L/(L + M). The intersections of the lines represent the achromatic point**.

## Parameter evaluation

To determine the influence of the model's parameters on the predicted color, simulations were conducted with different $SO's$ spatial parameters ($W$ and $\sigma$), $\alpha_i$ and $\alpha_c$. The simulation results over a CIELu'v' color space for both $\alpha_i$ and $\alpha_c$ (ranging from 0.5 to 1) are shown in **Fig 7**. With each image (colored face, cube with natural-yellowish-blueish illumination, #theDress, and #theShoe), we sampled pixels from different locations in the image where each location has a different color (hue) and ran simulations with varying $\alpha_i$ and $\alpha_c$. For further parameter evaluation, we also changed the kernel size of the $SO$ cell ($W = 21$, $\sigma = 5$, $W = 11$ and $\sigma = 3$; Eqs **4–6**). It appears that predictions can range over areas of CIELu'v' color space as well as curves. As

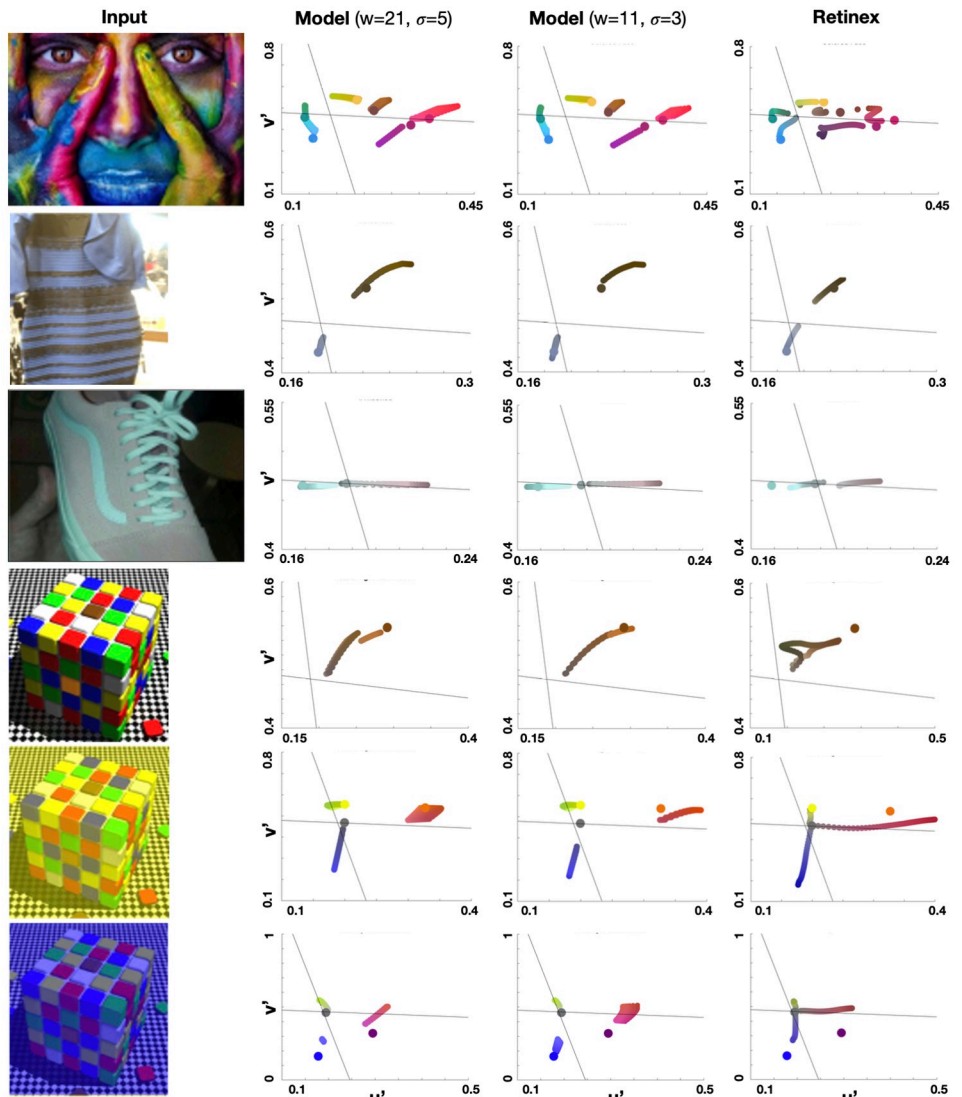

**Fig 7. Parameter evaluation with 21x21 and 11x11 kernel size (2<sup>nd</sup> and 3<sup>rd</sup> columns, respectively), compared with Retinex prediction (right column).** Input images are shown in the left column.

well, the model appears to be more sensitive to the selection $\alpha_i$ and $\alpha_c$ rather than it is to the selection of the spatial parameters of SO cells.

## Model comparison

We further compared the prediction of the model to a modified version of the Retinex algorithm, one of the most established retinal models in computational vision [51]. We used a single-scale non-logarithmic version of Retinex, as was suggested in [52]. Retinex predictions are described using:

$$Retinex(x, y, s) = I(x, y) - I * G(x, y, s). \tag{25}$$

where $0 \leq x \leq m - 1, 0 \leq y \leq n - 1$ where $m$ and $n$ are the image width and height respectively. In the Retinex algorithm, the filter response is computed separately for each of the three

image channels (R, G, B). Here, we changed the spatial scale of the Retinex predictions for a better alignment with our model, by modulating the parameter *s*:

$$G(x, y, s) = \frac{\exp\left(\frac{-(x^2+y^2)}{s^2}\right)}{\sum_{x=0}^{m-1} \sum_{y=0}^{n-1} \exp\left(\frac{-(x^2+y^2)}{s^2}\right)} \tag{26}$$

Small s values result in high pass responses, while large s values produce more low pass frequencies. As a result of the high-pass response, colors appear near the edges while achromatic areas appear between the edges. Therefore, the Retinex algorithm results are approaching the achromatic gray point when *s* decreases (**Fig 7, right column**). On the other hand, as *s* increases, Retinex's results obtain more chromatic colors (predicted colors that are closer to the original color, in CIELu'v' space). Retinex's results verify our model predictions regarding the individual perceptual differences in the images of #theDress and #theShoe (also demonstrated in [52]), as well as the perception of color under different illuminations. In contrast to Retinex, our proposed model is biologically plausible, allowing the attribution of these differences to the proportions between the single and double opponent cells' activity. Furthermore, in contrast to our model, Retinex was not able to predict as accurately the color assimilation effect (**S4 Fig**), showcasing the generality of our proposed computational framework.

## Discussion

Our parametric implementation of color perception allows critical evaluation of various visual phenomena in a single biological plausible computational framework. It uses a parametrized combination of high and low frequencies and an SNN-based filling-in process to provide adequate color image perception while accounting for individual perception differences. This work extends our previous SNN-based model [5], which addressed the images' intensity channels alone. We show that while in the perceptual reconstruction of natural color images, both single and double opponent pathways are required to achieve adequate results, the single opponent pathway is sufficient to predict the perception of the color assimilation grid illusion. Furthermore, we demonstrate individual differences in color perception using the #theDress and #theShoe images. Our proposed model can further explain both the watercolor [14] and the Cornsweet illusion [16] through the reconstruction of images from adapted gradients, as we recently demonstrated [32].

Our SNNs-driven computational framework follows the model suggested by Shapley and colleagues, which proposed dual opponent mechanisms for color perception [7]. When the color contrast is low, human color perception is characterized by spatially low pass filtering, where single-opponent neurons dominate visual perception. When color contrasts intensify, visual perception shifts from low pass to edge-sensitive filtering, where double-opponent neurons become the predominant mechanism. Our model parametrizes this duality with weighted channel contribution, allowing critical examination of the model's prediction. We modeled the single opponent pathway with low-pass filtering, implemented by convolving a Gaussian kernel- with an opponent color channel. The double opponent was modeled with high pass filtering, serving as color contrast detectors [53]. In our proposed model, rather than combine the double-opponent responses directly with the single-opponent responses, we used the double-opponent responses as triggers for diffusive Poisson-driven recurrent SNNs, allowing the reconstruction of low-pass properties from high-pass information. This is done in a diffusion-like process, in which a double-opponent cell activates its neighbors (Eq 16). Our recurrent SNN is a biologically plausible implementation of an iterative numerical solver of the Poisson equation, allowing accurate perceptual prediction. However, since an ensemble of spiking

neurons approximates each pixel's value, the process cannot reach a steady state, corresponding to the biological resource-constrained spike-based encoding.

In this work, we propose a biologically plausible SNNs-driven model which can serve as a potential neural mechanism for perceptual color filling-in, corresponding with the spreading of color signals in the cortex [3], [54]. Our model can be correlated with experimentational findings in the cortex, providing further insights. We show with voltage-sensitive dye-imaging in V1 of macaque monkeys in response to uniformly colored or achromatic large squares, that there is an unfilled area ('hole') of activity [42]. Our model predicts a similar pattern (partially filled square) when the chromatic alpha is large (**Fig 2**). It should be noted that layers 2–3 are the main cortical layers imaged by VSDI. The cells in these layers are mostly edge detectors and these layers contain a high population of double-opponent cells [6]. Therefore, to compare VSDI signals to simulations of black and red squares, we use $\alpha_i = 1$ and $\alpha_c = 1$ (large alphas) in our simulation to account for only the double opponent pathway. As we increase the single opponent pathway dominancy (by decreasing $\alpha$ and increasing $\beta$), this filling in the gap is shrinking, suggesting that the integration of the single and double opponent mechanisms does not occur in V1 but rather in higher visual regions. Furthermore, the cortex's layer hierarchy suggests that the receptive fields of higher visual layers correspond to wider spatial areas of the stimulus [1]. This can be interpreted as having recursive (horizontal connections) layer-based filling-in processes [55], which reduce the distance between edges in higher layers and the propagation time of the spreading filled-in signal [56]. Our model layer-based design supports this architecture. Consequently, this computational design can explain the results in [42], which conclude that while V1 activity is insufficient to explain the perception of filled objects, filling-in processes that occur at both low and high levels can produce the perception of filled objects. Experimental studies on filling-in are consistent with these ideas, as neural activities in V3 and V4 areas during perceptual filling-in effects were observed in response to the watercolor and Cornsweet illusions, texture, and afterimage filling-in [57–60].

Recently, Yang and colleagues demonstrated similar VSDI results [61]. Like the results reported here, they showed that V1 responses in cortical layers 2–3 are enhanced at the surface edges whereas the response at the surface center is suppressed. However, in this work we used a range of surface sizes, allowing us to compute the slope of population propagation from the edges to the surface's center. Thus, supporting our assumption of having horizontal connections contributing to the filling-in phenomenon.

We demonstrate the model's prediction of visual filling-in with various examples and critically examine related phenomena: color constancy, color assimilation, and individual perception. Color constancy and individual differences in color perception are widely discussed in the literature. For example, Dixon and Shapiro [52] suggested that these visual phenomena can be explained through high-pass filtering, which subtracts a blurred modality of an image from the original one and adds a constant intensity value, shifting it back into the viewable range [62]. This simple model was argued to account for different color perceptions, grounding it on individual frequency processing characteristics. Given the appropriate spatial parameters, the authors demonstrated that this model could explain color constancy in several illusions, such as the cube illusion [22] as well as the individual difference phenomenon regarding the #theShoe and #theDress (their results are also reproduced in this current work; **Fig 7**). However, this spatial filter cannot explain other brightness and color filling-in illusions, such as the Cornsweet [16] and the watercolor [14] illusions, as well as color assimilation (**S4 Fig**). A naïve frequency filtering cannot explain these illusions as they are based on changes generated at a thin edge, extending over large distances. While other models of color constancy do relays on the combinations of high and low-pass filters [13], [51], [63], they are not biologically plausible. First, they entail different filter kernel sizes to account for individuals' color

perceptions. Secondly, they require the original high-resolution image for processing, which is not conducted through the biological visual system. As was demonstrated by the model's prediction of #TheDress and #TheShoe, despite having only global parameters, our model was able to capture differences in perception with respect to the model's parameters (**Fig 7**). Thus, concurring with Gegenfurtner and colleagues, who found that there are multiple answers to the question, "what color is the dress?" [64]. Our results extend their finding and demonstrate in a biologically plausible computational framework that there can be multiple responses to any color image. In the colored face image, for example, people may perceive and name colors differently. Some may name the orange part of the woman's finger orange, while others might call it yellowish or greenish (the original orange color changes to greenish-yellowish with respect to the value of alpha; **Fig 7**, **top row**).

Our LPIPS-driven evaluation of the model demonstrates that LPIPS was unable to accurately capture illusive perception. Therefore, while the image with the lowest LPIPS score is the most perceptually similar to the GT, it might not capture the illusion. Therefore, here, LPIPS scores were used not to identify which parameters gave the lowest scores, but rather to illustrate: 1) the model can reconstruct images that are perceptually similar to GT; 2) that the demonstrated illusions, as were perceived by the brain, are different than the GT, resulting in a higher LPIPS score; and 3) that color perception varies across individuals. If someone perceives #theDress as black and blue, then their perception is more similar to the original/physical colors of the GT.

With #thedress image as a stimulus, numerous studies have attempted to identify the underlying mechanisms that lead to different perceptions of colors among individuals. Toscani and colleagues [65] investigated whether people who report different colors for #thedress do so because they have different assumptions about the illumination in the scene. They found that observers reporting the dress to be white (white perceivers) adjusted the background illumination more bluely than observers reporting it to be blue (blue perceivers). The illumination appeared less chromatic to blue perceivers. Therefore, they concluded that different assumptions about illumination chromaticity in the scene can explain ambiguity in the perceived color of the dress. Similarly, Witzel and colleagues [66] and Aston and colleagues [67] concluded that assumptions and priors about illumination affect perceived images. According to Witzel and colleagues [66], prior-modified images of the dress can manipulate the perceived color. The prior-modified image, however, did not predict the perceived color of the original dress image in all observers. Therefore, they concluded that interpretations of the dress' colors are influenced by assumptions about illumination, but other factors may systematically affect interpretations. Aston and colleagues [67] tested the possibility that color constancy could explain this phenomenon. A color constancy with illumination discrimination task was used to assess whether individual differences in generic color constancy could explain perception differences in our observers. Using the dress photograph as an example, they demonstrated that individual differences in perception may partly be explained by chromatic biases in illumination priors. Individual differences in color constancy, however, do not explain variability in the perception of dress colors. Observers individually discount achromatic features: while blue-black reporters focus on blue regions, while white-gold reporters focus on golden regions. It is consistent with the hypothesis that attention and local image statistics play a role in understanding multi-stable images. Overall, these studies confirmed the importance of background illumination, image statistics, and priors, but they could not explain the underlying mechanism. Our model allows the attribution of perceptual differences to the proportions between the single and double opponent cells' activity, while being general enough to account for a wide range of visual phenomena including color constancy, color assimilation, and ambiguous color perception.

In our suggested model, $\alpha_i$ and $\alpha_c$ are two parameters that can be modified to describe different visual phenomena (where $\beta_i = 1-\alpha_i$ and $\beta_c = 1-\alpha_c$). To demonstrate how the model can generate different perceptions, we manually modulated these parameters. In the brain, these weights, however, are determined by visual processes that were not computationally modeled. Moreover, these parameters might be computed locally within the brain, rather than globally, as in our current framework. The precise channels' weight setting is still unknown. When individual perceptual differences are considered (e.g., #theDress and #theShoe), $\alpha_i$ and $\alpha_c$ are vary among individuals and can even be different among images within an individual. For example, one person may perceive the dress as blue and black ($\alpha_c = 0.5$), and the shoe as white and pink ($\alpha_c = 1$). As a result, different images might be generated by a different weighting scheme. These parameters can be determined using an unknown set of visual features, supporting the concept of having them derived individually for each image. Therefore, we emphasize that our model does not suggest the existence of a single set of parameters with which all illusions can be accounted for, nor does it suggest that a single set of parameters would be sufficient for modeling individual color perception. We rather suggest that a weighted combination of the single opponent and the double opponent cells might model individual color perception. Interestingly, a recent work [50] showed that the perception of these visual imageries remains stable over time, possibly due to "one-shot learning," which allows for the first encounter's strong influence to determine observers' perception [50], [68]. Thus, suggesting that, once acquired, our color parameters stay constant. However, our model could be further developed to account for adaptive changes and the support of local alpha values, corresponding to the image's regional contrast differences.

## Supporting information

**S1 Fig. A comparison between our model prediction and the VSDI measurements across 0.5˚, 1˚, 2˚ and 3˚ profiles.**
(TIFF)

**S2 Fig. Reconstruction results without recurrent connections (i.e., without a filling-in network).**
(TIFF)

**S3 Fig. Reconstruction results with non-spiking and spiking neural networks.**
(TIFF)

**S4 Fig. Retinex prediction of the color assimilation effect, using filter size (s) in various sizes.**
(TIFF)

## Author Contributions

**Conceptualization:** Hadar Cohen-Duwek, Elishai Ezra Tsur.

**Data curation:** Hadar Cohen-Duwek, Hamutal Slovin, Elishai Ezra Tsur.

**Formal analysis:** Hadar Cohen-Duwek, Hamutal Slovin, Elishai Ezra Tsur.

**Funding acquisition:** Elishai Ezra Tsur.

**Investigation:** Hadar Cohen-Duwek, Elishai Ezra Tsur.

**Methodology:** Hadar Cohen-Duwek, Hamutal Slovin, Elishai Ezra Tsur.

**Project administration:** Elishai Ezra Tsur.

**Resources:** Hadar Cohen-Duwek, Elishai Ezra Tsur.

**Software:** Hadar Cohen-Duwek.

**Supervision:** Elishai Ezra Tsur.

**Validation:** Hadar Cohen-Duwek, Elishai Ezra Tsur.

**Visualization:** Hadar Cohen-Duwek, Hamutal Slovin, Elishai Ezra Tsur.

**Writing – original draft:** Hadar Cohen-Duwek, Hamutal Slovin, Elishai Ezra Tsur.

**Writing – review & editing:** Hadar Cohen-Duwek, Elishai Ezra Tsur.

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
