## [Decision Letter · Decision Letter 0]

6 Jun 2022

Dear Dr. Ezra Tsur,

Thank you very much for submitting your manuscript "Computational modeling of color perception with biologically plausible spiking neural networks" for consideration at PLOS Computational Biology.

As with all papers reviewed by the journal, your manuscript was reviewed by members of the editorial board and by several independent reviewers. In light of the reviews (below this email), we would like to invite the resubmission of a significantly-revised version that takes into account the reviewers' comments.

We cannot make any decision about publication until we have seen the revised manuscript and your response to the reviewers' comments. Your revised manuscript is also likely to be sent to reviewers for further evaluation.

Sincerely,

Tianming Yang

Associate Editor

PLOS Computational Biology

Wolfgang Einhäuser

Deputy Editor

PLOS Computational Biology

Reviewer's Responses to Questions

**Comments to the Authors:**

**Note that reviewer #3 has also uploaded an attachment.**

Reviewer #1: The authors of this paper constructed a spiking neural network model that is capable of reproducing several phenomena related to color perception. The model contained multiple biologically-plausible components, including low/high pass spatial filters and SNN-based filling-in processes. Overall, this is an interesting paper, but not without shortcomings. My major concerns are as follows:

1)While the proposed model consisted of multiple components, it’s unclear to me whether they are all useful. By assigning different weights to the single-opponent and double-opponent pathways, it could be seen they played different roles in different tasks. However, it’s still unclear whether SNN is necessary. More thorough investigations are needed for the readers to understand the specific contributions made by different components of the model.

2)While the model reproduced three phenomena related to color perception, the exact optimal parameters are different across contexts--the double-opponent component plays an important role in color constancy but not color assimilation. This inconsistency worries me. How does our brain determine which parameter should be used under certain situation? By arbitrarily assigning weights to the two components seems more like explaining different phenomena with different models...

3)Most of the analyses in the paper are qualitative in nature, making it difficult for the readers to judge the significance of this paper. In particular, as the authors mentioned, other models have also been proposed to account for color perception. I think it will be much more informative to also examine the performance of other baseline models on the same set of tasks, so we could better appreciate how this new model improves our understanding of color perception.

Minor:

1)The definition of the coefficients (α and β) in equation 17 seems to be inconsistent with elsewhere in the paper, e.g. Figure 1.

2)There are two Figure 4 in the figure legends.

Reviewer #2: Cohen-Duwek and colleagues propose a spiking neural network-based model to reconstruct perceptual color from retinal inputs. The model is image computable and has biologically relevant motifs (like single and double opponent cells, spatial filters etc.). This computational model was further compared against macaque V1 (voltage imaging) data and then used to explore three perceptual phenomena – color constancy, the color grid illusion, and ambiguous color perception. The authors’ claim in this paper is that this end-to-end model (model with both internal motifs like the brain and reconstruction performance like behavior) is a putative model of color perception in humans.

Overall, the paper addresses an interesting question about human color perception using a model which is simple and potentially interesting. I have several questions and issues with the paper that temper by enthusiasm. For instance, there are no statistical tests supporting any of the inferences (for example comparing models to the brain). I also feel the V1 data shown here provides a weak constraint to the model. There are no additional behavioral benchmarks to strengthen the paper – only self-reports by the authors.

Please find other major and minor points below.

Major points –

Comparison to V1 data. In Figure 1 the authors show the topographic map of neural activity in monkey V1 to black and red squares. The authors claim (without any supporting statistical tests) that simulation results correspond to the observed neuronal activations in V1. I have several questions about both the model and the inference drawn from this section which the authors must address.

1. How did the authors choose their parameters (alpha_i and alpha_c) decided? What happens for other beta and alpha values? The authors should show us the results for these choices that change over the course of the paper.

2. Why do the authors smooth the results instead of just showing us the average responses for all the model units (as in the original paper).

3. Do the authors feel that these data provide strong constraints on their specific model?

4. The authors should do more to quantify the match to data using appropriate statistical tests. They should also do some sensitivity analyses to show the dependence of their claim on the parameters of their model.

Color perception. Here the authors demonstrate the dependence of the model parameters on the perceived similarity (as judged by the LPIPS metric). They find that setting the parameters to 0.5 produces the lowest overall LPIPS over 4 representative images. I did not fully follow the logic of this section.

1. Was the goal of this simulation to find the model parameters that would give the lowers overall LPIPS score?

2. Given the model is image computable, why do the authors then not just repeat the simulations on many stimuli (say at least 100 or so)? That way we can be sure about the the result here which is that 0.5 is the current model parameter for a good image reconstruction. There is no motivation provided for using these 4 stimuli at all. I am sure the readers would be left wondering about the same question as well.

3. If the best model parameters are 0.5 for image reconstruction, how do the authors justify their original choice for Part 1 of the paper? (This is a recurring thing that I will bring up later as well).

Color constancy. The authors next go after, what I feel is the core problem, of color perception – color constancy. The claim here is that the model can factor out the effects of overall illumination and retrieve the true color of the rubiks cube – the model parameters for the best possible reconstruction is now a different number than used for the previous 2 sections.

The color assimilation grid illusion.

Here too the parameters change to a completely different number

Overall issue about the parameters.

The authors need to do more to convince the readers that the model parameter choice is not just double-dipping into the results. I can think of a few ways of doing this but let me suggest one.

For example, the authors could set the parameters based on the neural data (part 1 of the paper) and show the reconstruction results for the same parameters for the other experiments. This is the strongest way to do it – you are defining the internals of the model based on some (albeit weak) neural constraint and then testing reconstruction (perception/behavior) based on those parameters. I am afraid that the current claims would not hold if the authors, or would at least need to be tempered, if the authors choose to do this.

Minor points.

1. Introduction -- Color constancy is not an illusion

2. Typo – “as well as filtering” under the Color constancy section

Reviewer #3: Cohen-Duwek and colleagues implemented a modified SNN-driven model, with low and high spatial frequency filters for both chromatic and achromatic visual information, to explain both VSD imaging data measured in monkey V1 and the perception for color constancy, color assimilation and ambiguous colors. They demonstrated that such a single computational framework provides adequate color perception; and interestingly parameter changes in the model also explain individual differences for perceiving colors. Overall, the evaluation of critical model parameters for color perception, which suggested important roles for both single and double opponent channels, is very interesting; and the demonstration of how a SNN model combined with Poisson-processes to explain visually-driven responses in space and time is also important for further computational and perceptual studies. However, I do have comments and suggestions for revising the manuscript.

1) Model descriptions require more details and explanation. It is quite hard to follow the model without reading several times. I understand that the model is a modified one, which was built for accounting for neural responses driven by achromatic stimuli; but this doesn’t mean that some details of the model should be just referred to other paper. A), in the part for NEF, what type of computer language was used? The left side of the equation 1 is a function of x, but on the right side of the equation, there was no explicit x. Is the ai(x) in equation 2 the same as ai in equation 1? B) In the description of single and double opponent channels, spatial extent (sigma) of single opponent Gaussian spatial kernel was set at 5 pixels, but the operational matrix for double opponent channel (Laplacian operation) suggest its spatial extent is 1 pixel. The authors should give some explanations for why choosing spatial extents in this way. C), in the part of perceptual filling-in with spiking neurons, the feedback and tau in equation 13 were not explained. In this part, I suggest to change the subscripts for Is, because it is too similar to Ik or Ik-1, which has different meaning. In the last paragraph of this part, ‘Therefore, this connectivity scheme can be… as horizontal’ is incomplete. Please revise it. D) In the part of image perception, how is the function FI (for delta RG, delta BY and delta I) related to spikes in SNN model? A more explicit equation or explanation should be provided.

2) Results based on model simulation should have more discussions. The study used a model with SNN to explain VSD results. The author should have some discussion on the relationship between VSD signals and spiking activity in V1. The SNN model is a single layer network, which receives retinal input and simulates the neural activity in V1, presumably the superficial layer of V1 (V1 output layer). However, neurons in V1 output layer, in fact, mainly receive excitatory drives from V1 input layer, whose response properties might be different from those in the retina. I am wondering whether using different input for the model can generate different results for parameter alpha or beta. A recent study has demonstrated different spike activity patterns evoked by square stimuli at different V1 layers (Yang et al. 2022, Nature Communications). The author should discuss how their model will perform and whether their conclusion might be held, if the model receives input drive from V1 input layer shown in Yang et al. (2022).

3) Although several color related perceptions have been explored in this study, the conclusion for each result section should be written more clearly; and their relationship should be discussed. Can the parameter alpha from each section fit to an individual?

4) On page 6, it was written ‘Finally, the resulting surfaces were linearly combined with single opponent outputs to …’. How does this description fit to the later statement, on page 17, ‘rather than combine the ….’. I didn’t quite follow the ‘double-opponent responses as triggers for diffusive Poisson-driven recurrent SNNs’. Please adjust the corresponding method section to make this point more clear.

5) It is interesting to see that different alpha value for chromatic channel will lead to individual difference for perceiving the color of the dress. Gegenfurtner et al. 2015 (Current biology) should be discussed as well.

6) There were two figures for Figure 4 in the main text!

**Have the authors made all data and (if applicable) computational code underlying the findings in their manuscript fully available?**

Reviewer #1: Yes

Reviewer #2: **No: **I could not find any mention of this in the submitted paper.

Reviewer #3: **No: **Please make the code open to the public

PLOS authors have the option to publish the peer review history of their article (what does this mean?). If published, this will include your full peer review and any attached files.

Reviewer #1: No

Reviewer #2: No

Reviewer #3: No
---

## [Decision Letter · Decision Letter 1]

13 Aug 2022

Dear Dr. Ezra Tsur,

Thank you very much for submitting your manuscript "Computational modeling of color perception with biologically plausible spiking neural networks" for consideration at PLOS Computational Biology.

As with all papers reviewed by the journal, your manuscript was reviewed by members of the editorial board and by several independent reviewers. Although it was agreed that the revision was a significant improvement over the original manuscript, some of reviewer #1's comments were not satisfactorily addressed. Therefore, we would like to invite the resubmission of a significantly-revised version that takes into account reviewer #1's comments.

We cannot make any decision about publication until we have seen the revised manuscript and your response to the reviewers' comments. Your revised manuscript is also likely to be sent to reviewers for further evaluation.

Sincerely,

Tianming Yang

Associate Editor

PLOS Computational Biology

Wolfgang Einhäuser

Deputy Editor

PLOS Computational Biology

Reviewer's Responses to Questions

**Comments to the Authors:**

Reviewer #1: I appreciate that the authors included the Retinex model for direct comparison with their model, but my major concerns persist.

1)What happens when SNN is absent from the model, such as when the SO and DO channels are combined directly, or when a regular neural network is employed in its place?

2)What biological process controls how much weight is given to SO and DO channels? I think there should be some explanation for these weightings because they should be the outcome of biological processes, especially given the authors' emphasis on biological plausibility.

Reviewer #2: I appreciate the additional work put in by the authors. The quantitative comparisons with V1 data and additional simulation to show how things change as a function of the parameters are an important addition to the previous version of the paper. I still am not fully convinced about the authors' response to the choice of using parameters that are highly tunable. That being said, the new version of the paper makes this very clear which will hopefully also be appreciated by the readers. Therefore I am happy to recommend publication of the paper and congratulate the authors.

Reviewer #3: Thank the authors to fully address my comments. I have no further concern or comment.

**Have the authors made all data and (if applicable) computational code underlying the findings in their manuscript fully available?**

Reviewer #1: Yes

Reviewer #2: Yes

Reviewer #3: Yes

PLOS authors have the option to publish the peer review history of their article (what does this mean?). If published, this will include your full peer review and any attached files.

Reviewer #1: No

Reviewer #2: No

Reviewer #3: No
---

## [Decision Letter · Decision Letter 2]

10 Oct 2022

Dear Dr. Ezra Tsur,

We are pleased to inform you that your manuscript 'Computational modeling of color perception with biologically plausible spiking neural networks' has been provisionally accepted for publication in PLOS Computational Biology.

Best regards,

Tianming Yang

Academic Editor

PLOS Computational Biology

Wolfgang Einhäuser

Section Editor

PLOS Computational Biology

Reviewer's Responses to Questions

**Comments to the Authors:**

Reviewer #1: The authors have addressed all my concerns. I have no further questions.

Reviewer #2: I stay with my decision to accept the paper.

Reviewer #3: The authors have made great efforts to address reviewers' questions and revise the manuscript. I have no further question or concern.

**Have the authors made all data and (if applicable) computational code underlying the findings in their manuscript fully available?**

Reviewer #1: Yes

Reviewer #2: Yes

Reviewer #3: Yes

PLOS authors have the option to publish the peer review history of their article (what does this mean?). If published, this will include your full peer review and any attached files.

Reviewer #1: No

Reviewer #2: No

Reviewer #3: No

---

## [Editor Report · Acceptance letter]

22 Oct 2022

PCOMPBIOL-D-22-00565R2 

Computational modeling of color perception with biologically plausible spiking neural networks

Dear Dr Ezra Tsur,

I am pleased to inform you that your manuscript has been formally accepted for publication in PLOS Computational Biology. Your manuscript is now with our production department and you will be notified of the publication date in due course.

With kind regards,

Marianna Bach
